# Healthy and Unhealthy Plant-Based Diets and Glioma in the Chinese Population

**DOI:** 10.3390/brainsci13101401

**Published:** 2023-09-30

**Authors:** Weichunbai Zhang, Yue Peng, Xun Kang, Ce Wang, Feng Chen, Yongqi He, Wenbin Li

**Affiliations:** Department of Neuro-Oncology, Cancer Center, Beijing Tiantan Hospital, Capital Medical University, Beijing 100070, China; zwchunbai@163.com (W.Z.); pengyue227@163.com (Y.P.); kangxuntiantan@163.com (X.K.); 13146851017@163.com (C.W.); chenfeng@bjtth.org (F.C.); 122021010536@mail.ccmu.edu.cn (Y.H.)

**Keywords:** healthy plant-based diet, glioma, Chinese population, case–control study, unhealthy plant-based diet

## Abstract

Plant-based diets have been suggested to help prevent various chronic diseases, including cancer. However, there are few reports on central nervous system tumors, and data on dose–response relationships are lacking. This individual-matched case–control study included 506 cases and 506 controls. The overall plant-based diet index (PDI), the healthy plant-based diet index (hPDI), and the unhealthy plant-based diet index (uPDI) were calculated using dietary information collected through a food frequency questionnaire, with higher scores indicating better adherence. We analyzed the relationship of plant-based diets with glioma. After adequate adjustment for confounders, PDI was associated with a reduced glioma risk (OR = 0.42, 95% CI: 0.24–0.72). Conversely, uPDI was associated with an elevated glioma risk (OR = 8.04, 95% CI: 4.15–15.60). However, hPDI was not significantly associated with glioma risk (OR = 0.83, 95% CI: 0.48–1.45). For subgroups, PDI was not significant in analyzing young age, BMI, or any pathological subtypes. The restricted cubic spline function showed a significant dose–response relationship between PDI (*p_-nonlinearity_
*< 0.0001) and uPDI (*p_-nonlinearity_
*= 0.0711) and glioma. Further analysis found that refined grains had the greatest effect on gliomas in the less healthy plant-based food group. Therefore, following a plant-based diet was linked to a lower risk of glioma, especially when consuming fewer unhealthy plant-based foods.

## 1. Introduction

The main focus of a plant-based diet is plant foods, such as grains, vegetables, and fruits, etc., with the consumption of animal products being kept to a minimum [1,2]. This plant-based diet has long been thought to be effective in reducing overall mortality [3] and preventing and controlling many chronic diseases, such as cardiovascular disease [4], diabetes [5], and cancer [6]. Moreover, it maintained a similar attitude to several currently advocated dietary patterns, such as the DASH diet [7], with regard to certain food components. The EAT-Lancet report also recommended adhering to plant-based diets, which was not only conducive to personal health but also conducive to the sustainability of global resources [8]. 

Currently, a plant-based diet may provide protection against several types of cancer, including breast cancer [6], prostate cancer [9], and colorectal cancer [10]. However, there were few reports on central nervous system tumors, such as gliomas. Studies on the relationship between glioma risk and nutrition have mostly concentrated on dietary groups. Higher vegetable intakes significantly decreased glioma incidence (Relative risk (RR) = 0.78, 95% confidence interval (95% CI):0.69–0.87), according to a meta-analysis of 17 studies conducted by Ying et al. [11]. In contrast, a prospective cohort analysis of 545,770 participants by Dubrow et al. discovered no evidence of a substantial protective effect of vegetables on glioma (Hazard ratio (HR) = 1.17, 95% CI: 0.89–1.53) [12]. By examining the consumption of certain fruits such as watermelon (Odds ratio (OR) = 0.13, 95% CI: 0.02–1.00) and bananas (OR = 0.35, 95% CI: 0.18–0.71), earlier research in Canada also discovered that some fruits were connected to a lower risk of glioma [13]. However, according to a new meta-analysis, insufficient evidence of a link between fruits and gliomas was discovered [14]. Large cohort studies have also confirmed a link for some plant-based beverages, such as tea. In the United Kingdom Biobank cohort, Creed et al. studied the tea consumption of 379,259 individuals and found a negative correlation between tea consumption and the risk of glioma (RR = 0.69, 95% CI: 0.51–0.94) and tea appeared to be more protective against glioblastoma (RR = 0.55, 95% CI: 0.38–0.79) [15]. In addition, similar studies have been conducted on plant foods such as grains and nuts [14,16]. However, the relationship between these plant foods and glioma remained inconclusive.

On one hand, one of the potential mechanisms by which plant-based foods may prevent glioma or other cancers was the antioxidant [17,18,19]. While studying the relationship between food groups and glioma can help reveal potential biological mechanisms, it is important to note that different plant-based foods may have similar functions. Therefore, assessing a single food group may not accurately represent an individual’s overall plant-based diet. On the other hand, not all plant-based foods are beneficial to health, such as refined grains and sweets, etc. Although these are also plant foods, excessive consumption may still cause certain risks. When evaluating a plant-based diet, it is important to make proper distinctions. Therefore, in order to overcome the limitations of previous studies, this study introduced the diet index as a comprehensive measure of an individual’s plant-based food intake [4,5] and further divided it into the overall plant-based diet index (PDI), healthy plant-based diet index (hPDI), and unhealthy plant-based diet index (uPDI), to quantitatively evaluate the association with glioma of plant-based diets from different perspectives.

## 2. Materials and Methods

### 2.1. Study Population

This study design on diet and glioma has been previously reported [20]. In short, this case–control research involved in people over the age of 18. Among them, the case group was selected from glioma patients in Beijing Tiantan Hospital, via the convenience sampling method. Their diseases were diagnosed by professional doctors through imaging and pathological diagnostic data [21]. The control group consisted of healthy adults and was individually matched according to the same sex and age gap within five years. Based on this, this study did not include subjects who have ever had any type of cancer (except glioma) or hormone replacement therapy. Similarly, individuals with significant changes in eating behavior due to diseases of the digestive system, endocrine system, etc., or personal habits within the year before the survey were also excluded. All case and control subjects provided signed informed consent after understanding the purpose of the study. Finally, the analysis covered 1012 subjects altogether. The Institutional Review Board of Beijing Tiantan Hospital, Capital Medical University, approved the research protocol (No. KY2022-203-02).

### 2.2. Dietary Survey and Plant-Based Diet Index

To assess participants’ dietary intake over the past year, investigators collected dietary information face-to-face using a 114-item food frequency questionnaire (FFQ). The validity and reproducibility of the FFQ have been verified and described in detail in previous studies [20]. Subjects completed the frequency and intake of each item in the FFQ, and then it was determined how much of each food was consumed on average each day. The daily energy intake was calculated according to the energy information provided by the China food composition table [22].

Based on previous studies, we created three plant-based diet indices—PDI, hPDI, and uPDI [4]. These plant-based diet indices were assessed using data from eighteen different food groups. In short, these food groups were broadly divided into three categories. Healthy plant foods include legumes, vegetable oils, whole grains, fruits, nuts, vegetables, tea, and coffee. Less healthy plant foods include fruit juices, desserts, tubers, sugary drinks, and refined grains. Animal foods include animal fats, meat, dairy products, eggs, fish, seafood, and animal products. The daily intake of each food group was divided into quintiles and assigned 1–5 points. The direction of the assignment was determined according to the actual situation of the plant-based diet index. Positive scores were assigned as follows: the first quintile received 1 point, and the fifth quintile received 5 points. Negative scores were assigned as follows: the first quintile received 5 points, and the fifth quintile received 1 point. For the calculation of PDI, healthy and less healthy plant foods were assigned positive scores, while animal foods were assigned negative scores. For the calculation of hPDI, healthy plant foods were assigned positive scores, while less healthy plant foods and animal foods were assigned negative scores. For the calculation of uPDI, less healthy plant foods were assigned positive scores, while healthy plant foods and animal foods were assigned negative scores.

### 2.3. Covariates

Age, sex, income, smoking status, occupation, education level, allergy, head trauma, and family history of cancer were collected using questionnaires. Physical activity levels were estimated via a short questionnaire [23]. Height and weight were recorded through standard measurement methods, and body mass index (BMI) was assessed.

In addition, the presence of electromagnetic fields, such as broadcast antennas, near the home was defined as a high-risk area, which was also considered a potential confounding factor [24].

### 2.4. Statistical Analysis

All statistical analyses were performed via SPSS 26.0 and R 4.1.1. All statistical tests were two-sided, and *p* < 0.05 was considered statistically significant. The basic characteristics and dietary scores of the healthy population and glioma patients were compared by *t*-test and chi-square test. For the connection between plant-based diet indices and glioma, we used a logistic regression model to obtain ORs and 95% CIs and calculated the results for the plant-based diet indices as categorical and continuous variables, respectively. When the plant-based diet index was used as a categorical variable, it was converted into tertiles, with the first tertile as the reference group. Results for both categorical and continuous variables were adjusted for potential confounders, including disease history (allergies, head trauma, and family history of cancer), demographics (age, education, occupation, and family income), lifestyle (smoking, high-risk areas, and physical activity levels), BMI, and energy intake.

The subgroup analyses listed below were carried out to verify the stability of the results. The following subgroups were examined: different subgroups for age (≤40 and >40), sex (male and female), BMI (≤23.31 kg/m^2^ and >23.31 kg/m^2^), education level (middle school and below and university and above), household income (<CNY 3000/month and ≥CNY 3000/month), smoking status (never smoking and smoking), allergies (yes and no), or family history of cancer (yes and no).

To appropriately depict the glioma risk across the three plant-based diet indices, we utilized a restricted cubic spline function with four nodes situated at every 20 percentiles from the 20th percentile, with the reference set at the 10th percentile, and calculated a non-linear *p*-value by testing the second spline.

In addition, we further explored the association between each component of less healthy plant foods and glioma risk in an attempt to identify the major contributors to the association.

## 3. Results

### 3.1. Basic Characteristics of the Study Population

For the overall population, patients with glioma were mostly manual workers with slightly lower education levels, higher BMI, more physical activity, more smoking, and less likely to suffer from allergies but more likely to suffer from cancer in the family. There were also differences in economic income between the two groups. For study subjects with different BMI groups, patients with glioma performed more physical activity, and were more likely to suffer from cancer in the family. There were also differences in economic income between the two groups. For the low BMI group, the case group had a higher BMI (*p* = 0.007), a higher proportion of manual workers (*p* = 0.001), a lower education level (*p* < 0.001), a lower household income (*p* < 0.001), more smokers (*p* < 0.001), a lower proportion of allergic patients (*p* < 0.001), a higher proportion of family history of cancer (*p* = 0.038), and a lower proportion of low physical activity individuals (*p* < 0.001). For the high BMI group, the case group had a higher proportion of the female population (*p* = 0.008), a lower household income (*p* < 0.001), a higher proportion of family history of cancer (*p* = 0.013), and a lower proportion of low physical activity individuals (*p* < 0.001). See Table 1.

Figure 1 displays the three index scores for a plant-based diet. In terms of PDI, the cases were substantially lower than the controls (*p* = 0.003). In terms of uPDI, the cases were higher than the controls (*p* < 0.001). Between the two groups, there was no discernible change in hPDI (*p* = 0.418).

### 3.2. Association between Plant-Based Diet Indices and Glioma

Table 2 illustrates the relationship between glioma and the indices of plant-based diets. After controlling for covariates, the third tertile of the PDI was related to a reduced risk of glioma (OR = 0.42, 95% CI: 0.24–0.72), the third tertile of the uPDI was related to an elevated risk of glioma (OR = 8.04, 95% CI: 4.15–15.60), while the hPDI was not significantly related to the risk of glioma (OR = 0.83, 95% CI: 0.48–1.45). The outcomes of the continuous variables were consistent with those of categorical variables.

### 3.3. Plant-Based Diet Indices and Gliomas of Different Pathological Classifications and Grades

The results of glioma analysis according to different pathological classifications showed that uPDI was related to an elevated risk for astrocytoma and glioblastoma (astrocytoma: OR = 1.16, 95% CI: 1.04–1.31; glioblastoma: OR = 1.20, 95% CI: 1.10–1.31), while PDI and hPDI were not significantly related to either glioma subtype. The oligodendroglioma sample size was small, so no further analysis was made (Table 3).

According to the results of glioma analysis with different grades, uPDI was related to an elevated risk (OR = 1.11, 95% CI: 1.02–1.22) for low-grade gliomas. For high-grade gliomas, PDI was linked to a reduced risk (OR = 0.94, 95% CI: 0.89–0.99), and uPDI was linked to an elevated risk (OR = 1.15, 95% CI: 1.09–1.22). Others were not significantly related (Table 3).

### 3.4. Subgroup Analysis

Based on covariates, we performed subgroup analyses on the association between glioma risk and plant-based diet by excluding participants of different sexes, BMIs, ages, education level, family income, smoking status, allergies, and family history of cancer. The results showed that for sex subgroups, smoking status subgroups, BMI subgroups, family history of cancer subgroups, and other subgroups, uPDI was associated with an increased risk of glioma, while hPDI was not associated with glioma risk, which were consistent with the overall population results. However, for PDI, the results were not significant in the low age group, low BMI subgroup, low household income group, and allergic population, which was inconsistent with the overall population results and may be related to the small sample size of some subgroups (Appendix A).

### 3.5. Dose–Response Relationship

The restricted cubic spline function’s results revealed a dose–response relationship between glioma and the PDI that was significantly nonlinear (*p_-nonlinearity_
*< 0.0001), with an overall trend of first increasing and then decreasing, and the results were significant when the score exceeded 52. This dose–response relationship for the uPDI was significantly linear (*p_-nonlinearity_
*= 0.0711), with the risk of glioma increasing significantly with an uPDI above 38 points and leveling off after 52 points. There was no significant dose–response relationship between glioma and the hPDI (*p_-overall_
*= 0.9816). See Figure 2.

### 3.6. Association between Unhealthy Plant-Based Dietary Components and Glioma

As shown in Table 4, higher refined grains were related to an increased risk of glioma in the total population (OR = 1.94, 95% CI: 1.40–2.67), similar associations were found with glioblastoma (OR = 2.12, 95% CI: 1.19–3.78) and high-grade gliomas (OR = 2.61, 95% CI: 1.58–4.33), but no significant association was found with astrocytoma (OR = 1.43, 95% CI: 0.63–3.26) and low-grade glioma (OR = 1.27, 95% CI: 0.68–2.38). Higher tubers were related to a reduced risk of glioma in the general population (OR = 0.90, 95% CI: 0.86–0.95), and the results in different subtypes and grades were also consistent with the general population. However, the results for sweets and beverages were not significant (Table 4).

## 4. Discussion

This is the first time that the association between glioma and plant-based diets has been reported in the Chinese population. In this study on gliomas, we found that a higher PDI was related to a reduced risk, a higher uPDI was related to an elevated risk, and the hPDI was not related to the risk of glioma. Furthermore, it was found that the dose–response relationship of the PDI was nonlinear, while the dose–response relationship of the uPDI was linear. In addition, we conducted sufficient subgroup analysis based on relevant confounding variables. For the uPDI and hPDI, the results showed that the majority of subgroup results were consistent with the overall population, but the PDI results were not significant in individual subgroups, which was considered to be related to the small sample size of individual subgroups. However, the significance of the Model 1 and Model 2 results for the vast majority of subgroups was basically consistent. However, for the PDI, in the elderly subgroup and high BMI subgroup, Model 1 was significant while Model 2 was not, indicating that there may be potential confounding factors that may affect the results.

Unlike vegetarianism, plant-based diets promote the consumption of plant-based foods such as fruits and whole grains, while limiting animal products such as meat, eggs, and milk. Vegetarianism often bans all animal products or selectively consumes certain portions of animal products [25]. The plant-based diet index appeared to be more appropriate for assessing the overall plant food intake in most people’s diets. Several previous studies have reported a strong correlation between tumors and plant-based diets. Kane-Diallo et al. followed 42,544 participants over 45 years of age for an average of 4.3 years and found that higher plant-based diet scores were inversely linked to total cancer risk (HR = 0.85, 95% CI: 0.76–0.97), particularly gastrointestinal cancers (HR = 0.68, 95% CI: 0.47–0.99) and lung cancer (HR = 0.47, 95% CI: 0.25–0.90) [26]. Sasanfar et al. also found a significant correlation between breast cancer and higher hPDI scores (OR = 0.55, 95% CI: 0.37–0.82) [6]. Wu et al. conducted a comprehensive study of the Chinese population and found that adherence to plant-based diets (OR = 0.79, 95% CI: 0.66–0.95), especially healthy plant-based diets (OR = 0.45, 95% CI: 0.38–0.55), was associated with a reduced risk of colorectal cancer [10]. Despite this, studies on neuro-oncology are rare. So far, only Mousavi et al. reported that participants with higher scores on the PDI (OR = 0.54, 95% CI: 0.32–0.91) and the hPDI (OR: 0.32, 95% CI: 0.18–0.57) in the Iranian population had significantly lower rates of glioma [27]. This broadly matched what we discovered. Our study found that higher PDI was associated with a reduced glioma risk (OR = 0.42, 95% CI: 0.24–0.72), and similar results were also observed in the high-grade glioma population (OR = 0.94, 95% CI: 0.89–0.99). The dose–response suggested an inverted U-shaped trend in the relationship between the PDI and glioma, but the protective effect of plant-based diets was significant only after 52 points. The biological mechanism of plant-based diets affecting the development of glioma is not clear, and previous studies have proposed several plausible hypotheses. On the one hand, as the name suggested, the plant-based diet is characterized by a large intake of plant-based foods rich in various phytochemicals, such as carotenes, flavonoids, and anthocyanins. In preclinical studies, the positive effects of these phytochemicals on gliomas have been found. According to research by Coelho et al., the flavonoid apigenin inhibited the ability of rat C6 glioma cells to survive, grow, proliferate, and migrate. This was achieved by inducing differentiation, apoptosis, and autophagy, as well as participating in regulating immune responses [28]. Anthocyanins can also exert antiproliferative effects by regulating the glycolytic metabolism and protein levels associated with the survival of glioma cells [29]. These phytochemicals had different mechanisms of action and may affect glioma development in an additive or synergistic way. Moreover, plant-based foods were also abundant in micronutrients such as vitamins and minerals. These antioxidant nutrients were found to have an association with a reduced glioma risk [20,30]. On the other hand, it may be related to insulin-like growth factor 1 (IGF-1) [31,32]. IGF-1 was a growth factor in a variety of malignant cells, and existing studies have confirmed that high circulating levels of IGF-1 increase cancer risk [32,33,34], and gliomas are no exception [35,36]. The plant diet has a significant regulatory effect on the IGF-1 levels in the body [37]. In addition, it might also be correlated with limiting the intake of animal foods. Red or processed meat includes N-nitroso or heme iron, both of which have carcinogenic effects [38,39].

For the uPDI, higher scores were associated with elevated risks of colorectal cancer [10] and breast cancer [40]. Our study also observed a similar association (OR = 8.04, 95% CI: 4.15–15.60). This association remained relatively stable, as the same results were discovered in subgroups with various types and grades. Further, there was a significant linear dose–response relationship between the uPDI and glioma. Previously, however, the same association was reported only in studies of diet and glioma based on the Iranian population (OR = 2.85, 95% CI: 1.26–6.47) [27]. The uPDI index was mainly characterized by a higher intake of refined grains, sweets, etc. Although these foods were also plant-based, they contained relatively high amounts of carbohydrates, added sugars, or artificial sweeteners. These components may lead to changes in the gut microbiota [41], which may impact brain tumors, including gliomas, by the brain–gut axis [42,43]. Furthermore, considering the Warburg effect, glioma cell metabolism may be more inclined to glycolysis [44]. Ketogenic diets, characterized by low carbohydrates, may reduce tumor growth [45]. The Mediterranean diet, DASH diet, etc., also limited these unhealthy plant foods. These dietary patterns have been reported to have a protective effect on glioma [46,47].

Unlike other studies, our study observed significant correlations between the PDI and uPDI and glioma. However, the results for the hPDI were not significant. Although this was inconsistent with the results of Mousavi et al., we believed that this can be reasonably explained [27]. First, the results suggested that in addition to healthy plant foods, some unhealthy plant food components may also have a protective effect against glioma. Therefore, in further analysis, we found that higher tubers intake was linked to a reduced risk of glioma (OR = 0.90, 95% CI: 0.86–0.95). Because the PDI was originally used to explore the correlation between diabetes and diets [5], starch-rich tubers were listed as unhealthy plant foods. However, nutritional epidemiological studies have showed that potato intakes are associated with cancer mortality [48] and certain cancers [49]. Second, consider eating habits. There was a large difference in tubers consumption between the two studies’ participants, especially in the healthy control group; the tuber consumption of the Chinese population was about twice that of the Iranian population (Chinese population: 46.43 ± 58.95 g/d, Iranian population: 20.3 ± 20.0 g/d) [27]. As for the potential risk of the uPDI, in addition to the lower intake of healthy plant foods, since the Chinese population has a lower intake of sweets and desserts, the main reason was the intake of refined grains, which was also consistent with previous results [50]. In addition, tubers such as potatoes are rich in chlorogenic acid [51], which has been found to have potential therapeutic effects on glioma [52]. This may also lead to tubers being a positive food for glioma, rather than a negative one.

This study had some limitations. First, this study was a case–control study, and it is important to note that there may be inherent recall bias and selection bias associated with this study design, which may not be entirely avoidable. However, due to the extremely low incidence of glioma, case–control studies are still the primary type of study for investigating the causes of glioma. Therefore, we look forward to using prospective cohort studies in future studies to further validate the relationship between plant-based diets and glioma. We also minimized some bias during the study by assisting participants in assessing food intake through dietary mapping, using validated food frequency questionnaires, and selecting patients with newly diagnosed glioma. Secondly, the glioma patients in this study were mainly from Beijing Tiantan Hospital, which may introduce some selection bias. However, Beijing Tiantan Hospital is currently the most authoritative medical institution in the field of neuro-oncology in China. Its patients also come from all over the country, so it is reasonably representative of the Chinese population. Considering that the sample was only from the Chinese population, its representativeness was limited, and its conclusions should be extrapolated to other populations with caution. In addition, due to the complex composition of plant-based diets, the vast majority of research on tumors has mainly focused on population studies. Therefore, the anti-tumor mechanism of plant-based diets is not yet clear, and further exploration is needed in future studies. Despite this, the study had some advantages. First, it was the first study on the association between plant-based diets and glioma in the Chinese population. Considering the differences between the Chinese and Western diets and the representation of the Asian region, the results have significant reference value for regions with similar diet structures. Secondly, this study further investigated the association between plant-based diets and different pathological classifications and grades, especially fully exploring the dose–response relationship between plant-based diets and glioma risk, which has not been reported in previous studies.

Overall, our study found that adherence to a plant-based diet and avoidance of unhealthy plant-based diets were associated with a reduced risk of glioma. In the future, this finding should be validated by prospective studies and its underlying mechanisms explored.

## Figures and Tables

**Figure 1 brainsci-13-01401-f001:**
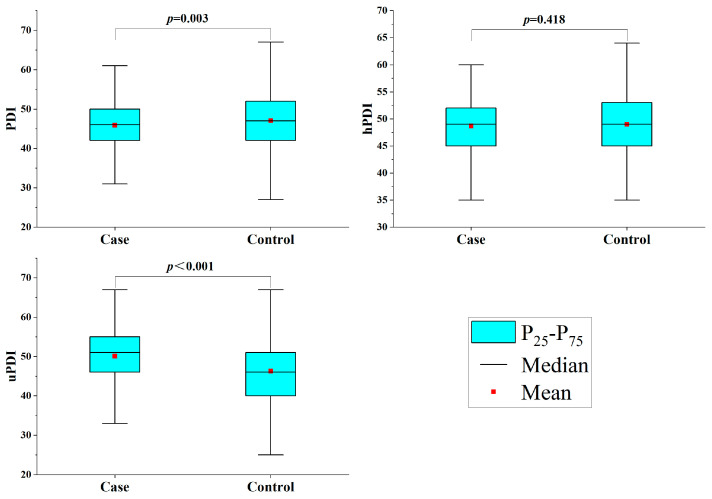
Three plant-based dietary indices of the study subjects.

**Figure 2 brainsci-13-01401-f002:**
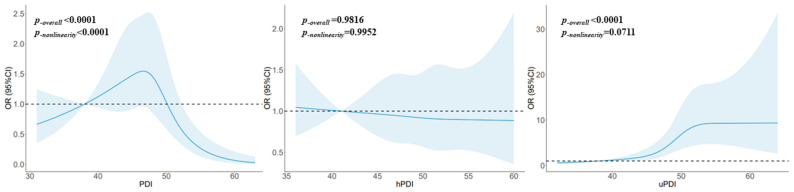
The restricted cubic spline for the associations between plant-based diet indices and glioma. The lines represent adjusted ORs based on restricted cubic splines for the intake in the regression model. Knots were placed at the 20th, 40th, 60th, and 80th percentiles of the plant-based diet indices, and the reference value was set at the 10th percentile. The adjusted factors were the same as in Model 2.

**Table 1 brainsci-13-01401-t001:** Basic characteristics of the study population.

	BMI ≤ 23.31 kg/m^2^			BMI > 23.31 kg/m^2^			*p* ^b^
	Case (*n* = 223)	Control (*n* = 283)	*p* ^a^	Case (*n* = 283)	Control (*n* = 223)	*p* ^a^	
Age(years)	41.77 ± 13.18	37.79 ± 11.58	<0.001	43.29 ± 13.01	45.41 ± 13.14	0.070	0.072
Sex (%)			0.275			0.008	1.000
Male	46.2	41.3		64.0	74.9		
Female	53.8	58.7		36.0	25.1		
BMI (kg/m^2^)	21.14 ± 1.66	20.73 ± 1.71	0.007	26.31 ± 2.22	26.00 ± 2.24	0.125	<0.001
High-risk residential area (%)			0.873			0.450	0.534
Yes	21.1	20.5		21.6	18.8		
No	78.9	79.5		78.4	81.2		
Occupation (%)			0.001			0.675	0.024
Manual workers	28.3	14.8		25.1	27.4		
Mental workers	52.0	66.1		52.6	53.3		
Others	19.7	19.1		22.3	19.3		
Education level (%)			<0.001			0.062	<0.001
Primary school and below	6.7	1.8		7.1	3.6		
Middle school	42.2	17.0		41.0	35.4		
University and above	51.1	81.2		51.9	61.0		
Household income (%)			<0.001			<0.001	<0.001
<CNY 3000/month	9.9	15.9		9.5	21.1		
CNY 3000–10,000/month	78.0	48.4		74.2	50.2		
>CNY 10,000/month	12.1	35.7		16.3	28.7		
Smoking status (%)			<0.001			0.197	0.039
Never smoking	70.8	85.5		69.2	62.3		
Former smoking	12.6	3.5		13.1	13.9		
Current smoking	16.6	11.0		17.7	23.8		
History of allergies (%)			<0.001			0.124	<0.001
Yes	4.9	14.8		9.9	14.3		
No	95.1	85.2		90.1	85.7		
History of head trauma (%)			0.930			0.499	0.474
Yes	9.4	9.2		12.7	10.8		
No	90.6	90.8		87.3	89.2		
Family history of cancer (%)			0.038			0.013	0.001
Yes	29.6	21.6		30.4	20.6		
No	70.4	78.4		69.6	79.4		
Physical Activity (%)			<0.001			<0.001	<0.001
Low	14.8	47.3		12.7	43.9		
Moderate	36.8	35.7		44.9	37.2		
Violent	48.4	17.0		42.4	18.9		

^a^ *p*-values were derived from Student’s *t*-tests for continuous variables according to the data distribution and the chi-square test for the classified variables. ^b^ Results of the overall case group and the overall control group.

**Table 2 brainsci-13-01401-t002:** Association between plant diet indices and glioma.

	T1	T2	T3	Continuous ^c^	*p*
PDI	<45	45–49	>49		
Case/Control	205/194	172/127	129/185		
Model 1 ^a^	1	1.27 (0.94–1.72)	0.66 (0.48–0.89)	0.97 (0.96–0.99)	0.004
Model 2 ^b^	1	1.52 (0.91–2.51)	0.42 (0.24–0.72)	0.94 (0.91–0.98)	0.002
hPDI	<48	48–52	>52		
Case/Control	202/195	181/165	123/146		
Model 1 ^a^	1	1.03 (0.78–1.37)	0.80 (0.57–1.10)	0.99 (0.97–1.01)	0.398
Model 2 ^b^	1	0.73 (0.46–1.18)	0.83 (0.48–1.45)	0.99 (0.96–1.03)	0.685
uPDI	<46	46–52	>52		
Case/Control	119/247	189/148	198/111		
Model 1 ^a^	1	2.85 (2.03–3.99)	4.11 (2.87–5.89)	1.08 (1.06–1.10)	<0.001
Model 2 ^b^	1	4.60 (2.57–8.23)	8.04 (4.15–15.60)	1.13 (1.09–1.18)	<0.001

^a^ Model 1: unadjusted model through logistic regression. ^b^ Model 2: adjusted for age, BMI, occupation, education level, household income, high-risk residential areas, smoking status, history of allergies, history of head trauma, family history of cancer, physical activity, and energy intake. ^c^ PDI, hPDI, and uPDI per 1-point increments.

**Table 3 brainsci-13-01401-t003:** Association between plant-based diet indices and different pathological classifications of glioma.

Pathological Subgroup ^c^	Model 1 ^a^	*p*	Model 2 ^b^	*p*
Pathological classification				
Astrocytoma				
PDI	0.98 (0.94–1.02)	0.322	0.99 (0.90–1.10)	0.927
hPDI	1.02 (0.97–1.07)	0.533	1.06 (0.94–1.20)	0.334
uPDI	1.05 (1.01–1.10)	0.009	1.16 (1.04–1.31)	0.011
Glioblastoma				
PDI	0.98 (0.95–1.01)	0.153	0.95 (0.89–1.01)	0.097
hPDI	0.98 (0.95–1.02)	0.302	1.02 (0.96–1.09)	0.563
uPDI	1.10 (1.07–1.14)	<0.001	1.20 (1.10–1.31)	<0.001
Pathological grade				
Low grade				
PDI	0.99 (0.95–1.04)	0.778	0.97 (0.89–1.06)	0.535
hPDI	1.05 (0.99–1.11)	0.101	1.05 (0.96–1.16)	0.300
uPDI	1.06 (1.02–1.10)	0.005	1.11 (1.02–1.22)	0.020
High grade				
PDI	0.97 (0.95–0.99)	0.014	0.94 (0.89–0.99)	0.014
hPDI	0.98 (0.95–1.01)	0.192	0.99 (0.94–1.05)	0.800
uPDI	1.07 (1.05–1.10)	<0.001	1.15 (1.09–1.22)	<0.001

^a^ Model 1: unadjusted model through logistic regression. ^b^ Model 2: adjusted for age, BMI, occupation, education level, household income, high-risk residential areas, smoking status, history of allergies, history of head trauma, family history of cancer, physical activity, and energy intake. ^c^ PDI, hPDI, and uPDI per 1-point increments.

**Table 4 brainsci-13-01401-t004:** Association between unhealthy plant-based dietary components and glioma.

Food Groups ^c^	Refined Grains	Tubers	Desserts	Sugary Drinks
Total population				
Model 1 ^a^	1.85 (1.57–2.17)	0.91 (0.88–0.95)	0.99 (0.96–1.03)	1.00 (0.99–1.01)
Model 2 ^b^	1.94 (1.40–2.67)	0.90 (0.86–0.95)	1.02 (0.95–1.08)	1.00 (0.99–1.01)
Astrocytoma				
Model 1 ^a^	1.66 (1.17–2.35)	0.88 (0.81–0.95)	0.99 (0.93–1.06)	1.01 (0.99–1.02)
Model 2 ^b^	1.43 (0.63–3.26)	0.82 (0.70–0.96)	1.08 (0.94–1.24)	1.01 (0.99–1.04)
Glioblastoma				
Model 1 ^a^	1.92 (1.51–2.43)	0.90 (0.86–0.95)	0.97 (0.92–1.03)	1.01 (0.99–1.02)
Model 2 ^b^	2.12 (1.19–3.78)	0.78 (0.68–0.90)	0.97 (0.84–1.12)	1.00 (0.98–1.02)
Low grade				
Model 1 ^a^	1.61 (1.17–2.22)	0.85 (0.77–0.95)	1.01 (0.94–1.09)	0.98 (0.96–1.01)
Model 2 ^b^	1.27 (0.68–2.38)	0.71 (0.56–0.91)	1.14 (0.96–1.36)	0.97 (0.93–1.01)
High grade				
Model 1 ^a^	1.83 (1.50–2.24)	0.90 (0.86–0.94)	0.99 (0.95–1.03)	1.01 (0.99–1.02)
Model 2 ^b^	2.61 (1.58–4.33)	0.87 (0.80–0.95)	0.97 (0.88–1.07)	1.01 (0.99–1.03)

^a^ Model 1: unadjusted model through logistic regression. ^b^ Model 2: adjusted for age, BMI, occupation, education level, household income, high-risk residential areas, smoking status, history of allergies, history of head trauma, family history of cancer, physical activity, and energy intake. ^c^ Refined grains per 100 g/d increments; tubers and desserts per 10 g/d increments; sugary drinks per 10 mL/d increment.

## Data Availability

The data presented in this study are available on request from the corresponding author.

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
