# Peer review of "Healthy and Unhealthy Plant-Based Diets and Glioma in the Chinese Population"

_brainsci, 2023, doi:10.3390/brainsci13101401_

Round 1

Reviewer 1 Report

1.   Please analyze and compare the glioma risk of hPDI and uPDI diets population between male and female, between low BMI (<23.31 and high BMI (>23.31) as well as display the data of statistical analysis.

2.   Please analyze and compare the glioma risk of hPDI and uPDI diets population among different smoking status.

3.   Whether family history of caner has different glioma risk in between hPDI and uPDI diets population.  

Author Response

Dear Reviewer

The authors would like to express their gratitude for the positive and constructive comments and suggestions. We have revised the manuscript and would like to resubmit it for your consideration. The modifications were marked are marked in red font in this manuscript. Here are the point-by-point responses:

1.Please analyze and compare the glioma risk of hPDI and uPDI diets population between male and female, between low BMI (<23.31 and high BMI (>23.31) as well as display the data of statistical analysis.

=> Responses: Thank you for your suggestion. We have supplemented this section. For populations of different sex, higher PDI was associated with a reduced risk of glioma, higher uPDI was associated with an increased risk of glioma, while hPDI was not associated with glioma. For populations with different BMI, higher uPDI was associated with an increased risk of glioma, while hPDI is not associated with glioma. However, for PDI, it was associated with a reduced risk of glioma in the high BMI subgroup, but not significantly in the low BMI subgroup. Please refer to Table S1 in the supplementary materials.

2.Please analyze and compare the glioma risk of hPDI and uPDI diets population among different smoking status.

=> Responses: Thank you for your suggestion. We have supplemented this section. For populations with different smoking statuses, higher PDI was associated with a reduced risk of glioma, higher uPDI was associated with an increased risk of glioma, while hPDI was not associated with glioma. Please refer to Table S1 in the supplementary materials.

3. Whether family history of caner has different glioma risk in betweenhPDI and uPDI diets population.  

=> Responses: Thank you for your suggestion. We have supplemented this section. For populations with different family history of caner, higher PDI was associated with a reduced risk of glioma, higher uPDI was associated with an increased risk of glioma, while hPDI was not associated with glioma. Please refer to Table S1 in supplementary materials.

Reviewer 2 Report

This study examined the relationship between plant-based diets and glioma risk in the Chinese population. The study found that adherence to a plant-based diet was associated with a reduced risk of glioma, while an unhealthy plant-based diet increased the risk. Notably, refined grains appeared to have a significant impact on glioma risk within the unhealthy plant-based food group. These findings suggest that dietary choices may play a role in glioma development. However, further research is needed to confirm these associations and explore underlying mechanisms.

Needs little improvements. 

Author Response

Dear Reviewer

The authors would like to express their gratitude for the positive and constructive comments and suggestions. We have revised the manuscript and would like to resubmit it for your consideration. The modifications were marked are marked in red font in this manuscript. Here are the point-by-point responses:

This study examined the relationship between plant-based diets and glioma risk in the Chinese population. The study found that adherence to a plant-based diet was associated with a reduced risk of glioma, while an unhealthy plant-based diet increased the risk. Notably, refined grains appeared to have a significant impact on glioma risk within the unhealthy plant-based food group. These findings suggest that dietary choices may play a role in glioma development. However, further research is needed to confirm these associations and explore underlying mechanisms.

=> Responses: Thank you for your suggestion. This is a very good suggestion! Our study actually focused on population epidemiology, exploring for the first time the association between plant-based diet and glioma in the Chinese population, in order to identify risk factors for glioma from a dietary perspective and provide a basis for primary prevention of glioma. However, as plant-based diet was a broad concept, it was difficult to conduct experiments like drugs or other intervention factors. Therefore, its related mechanisms need further exploration. However, our research team has conducted research on the mechanism of related phytochemicals in the intervention and treatment of brain tumors based on plant-based diets [1-5], providing some ideas for this. We have added this section to the limitations of the discussion. Please refer to Line 341-344 in the manuscript.

“In addition, due to the complex composition of plant-based diets, the vast majority of research on tumors was mainly focused on population studies. Therefore, the an-ti-tumor mechanism of plant-based diets was not yet clear, and further exploration was needed in future studies.”

  1. Kang, Z.; Li, S.; Kang, X.; etc. Phase I study of chlorogenic acid injection for recurrent high-grade glioma with long-term follow-up. Cancer Biol Med 2023, 20, 465-476, doi:10.20892/j.issn.2095-3941.2022.0762.
  2. Wang, C.; Cai, Z.; Huang, Y.; etc. Honokiol in glioblastoma recurrence: a case report. Front Neurol 2023, 14, 1172860, doi:10.3389/fneur.2023.1172860.
  3. Li, S.; Chen, J.; Fan, Y.; etc. Liposomal Honokiol induces ROS-mediated apoptosis via regulation of ERK/p38-MAPK signaling and autophagic inhibition in human medulloblastoma. Signal Transduct Tar 2022, 7, 49, doi:10.1038/s41392-021-00869-w.
  4. Li, S.; Li, L.; Chen, J.; etc. Liposomal honokiol inhibits glioblastoma growth through regulating macrophage polarization. Ann Transl Med 2021, 9, 1644, doi:10.21037/atm-21-1836.
  5. Zhang, W.; Wang, C.; Chen, F.; etc. Phytochemicals and Glioma: Results from Dietary Mixed Exposure. Brain Sci 2023, 13, doi:10.3390/brainsci13060902.

Reviewer 3 Report

This paper reported the relationship between dietary factor and glioma. There were not much published data in this area, and considering of the relatively large sample included, this paper should be valuable to the area. However, some issues need to be solved.

1. The results need re-phrasing. Readers don't want to see a plain transcription of the table 1. Please highlight the key difference/similarity related with glioma. 

2.  I'm quite confused to read table 3. Are these results from the logistic model, or a different analysis. If it was chi-square analysis for Model 1, please check all the ORs if there're correct.

3. Discussion was not done well, either. Line 297 says "Although this was contrary to the results of Mousavi et al., we thought it was consistent". This kind of statement is confusing. Please refer to English Language comments for more details and seek for academic English writing helps.

The English writing of this paper must be improved. Many grammar issues were noticed, some sentences were not complete or hard to understand. Examples include line 62-64, line 101-108. The use of connectives is not precise, either. 

Author Response

Dear Reviewer

The authors would like to express their gratitude for the positive and constructive comments and suggestions. We have revised the manuscript and would like to resubmit it for your consideration. The modifications were marked are marked in red font in this manuscript. Here are the point-by-point responses:

Reviewer 3

This paper reported the relationship between dietary factor and glioma. There were not much published data in this area, and considering of the relatively large sample included, this paper should be valuable to the area. However, some issues need to be solved.

  1. The results need re-phrasing. Readers don't want to see a plain transcription of the table 1. Please highlight the key difference/similarity related with glioma. 

=> Responses: Thank you for your suggestion. We have rewritten this section. Please refer to Line 154-161 in the manuscript.

“For the low BMI group, the case group had a higher BMI (P=0.007), a higher proportion of manual workers (P=0.001), a lower education level (P<0.001), a lower household income (P<0.001), more smokers (P<0.001), a lower proportion of allergic patients (P<0.001), a higher proportion of family history of cancer (P=0.038), and a higher proportion of low physical activity individuals (P<0.001). For the high BMI group, the case group had a higher proportion of female population (P=0.008), lower household income (P<0.001), higher proportion of family history of cancer (P=0.013), and higher proportion of low physical activity individuals (P<0.001).”

2.I'm quite confused to read table 3. Are these results from the logistic model, or a different analysis. If it was chi-square analysis for Model 1, please check all the ORs if there're correct.

=> Responses: Thank you for your suggestion. Model 1 and Model 2 in Table 3 were both derived from logistic regression results, with Model 1 not adjusting for any confounding factors. Table 2 adjusted for potential confounding factors for gliomas. We have modified the expression of the table to make the results presented clearer. Please refer to Line 181, Line 199, and Line 237 in the manuscript.

  1. Discussion was not done well, either. Line 297 says "Although this was contrary to the results of Mousavi et al., we thought it was consistent". This kind of statement is confusing. Please refer to English Language comments for more details and seek for academic English writing helps.

=> Responses: Thank you for your suggestion. I apologize for our unclear expression. We have modified this section. In addition, we have had native English speakers revise the English grammar of the manuscript. Please refer to Line 309-311 in the manuscript.

“Although this was inconsistent with the results of Mousavi et al., we believed that this can be reasonably explained.”

  1. The English writing of this paper must be improved. Many grammar issues were noticed, some sentences were not complete or hard to understand. Examples include line 62-64, line 101-108. The use of connectives is not precise, either.

=> Responses: Thank you for your suggestion. I apologize for our unclear expression. we have had native English speakers revise the English grammar of the manuscript. Please refer to Line 62-65 and Line 102-111 in the manuscript.

“On the other hand, not all plant-based foods were beneficial to health, such as refined grains and sweets, etc. Although also plant foods, excessive consumption may still cause certain risks. When evaluating a plant-based diet, it is important to make proper distinctions.”

“The direction of the assignment was determined according to the actual situation of the plant-based diet index. Positive scores were assigned as follows: the first quintile received 1 point, and the fifth quintile received 5 points. Negative scores were assigned as follows: the first quintile received 5 points, and the fifth quintile received 1 point. For the calculation of PDI, healthy and less healthy plant foods were assigned positive scores, while animal foods were assigned negative scores. For the calculation of hPDI, healthy plant foods were assigned positive scores, while less healthy plant foods and animal foods were assigned negative scores. For the calculation of uPDI, less healthy plant foods were assigned positive scores, while healthy plant foods and animal foods were assigned negative scores.”

Reviewer 4 Report

This study presents intriguing findings on the relationship between plant-based diets and glioma risk in the Chinese population. However, methodological concerns and potential biases limit the generalizability and impact of the results. My concerns and suggestions are outlined below:

·        The study employs a case-control design which, while appropriate for rare diseases, inherently has limitations. Such a study design is susceptible to both recall and selection biases. The authors acknowledged this limitation, but the possibility of unaddressed confounding factors remains a concern.

·        Using a convenience sampling method for selecting the case group raises concerns about the representativeness of the sample. Such a method can introduce bias and limit the external validity of the findings.

·        While Beijing Tiantan Hospital may be authoritative in the field of neuro-oncology in China, sourcing all glioma patients from a single hospital can limit the generalizability of the study findings.

·        Relying solely on food frequency questionnaires (FFQs) can lead to errors in dietary recall and may not capture the intricacies of participants' diets. Moreover, FFQ validation is mentioned, but details on the process or its accuracy are lacking.

·        Although the study controlled for some potential confounders, it did not account for others that could be crucial, such as genetic factors, environmental exposures, or other lifestyle factors that may influence glioma risk.

·        The authors make broad claims regarding the association between plant-based diets and glioma risk based on a sample exclusively from the Chinese population. Generalizing these findings to other populations can be misleading.

·        The lack of significant association between the healthy plant-based diet index (hPDI) and glioma risk is puzzling, especially given the observed effects of the overall PDI and uPDI. This inconsistency is not adequately addressed or explained.

·        The protective effects of tubers are highlighted, but it's unclear why they were initially classified as unhealthy plant foods in the PDI. This discrepancy necessitates clearer justification.

·        The paper frequently compares its findings to those from studies in other populations, particularly Iranian studies. While drawing parallels can be informative, the paper should stand on its own merit.

·        While potential mechanisms linking plant-based diets and glioma are proposed, they are speculative. A more thorough exploration and stronger evidence are required before suggesting any causative relationship.

Author Response

Dear Reviewer

The authors would like to express their gratitude for the positive and constructive comments and suggestions. We have revised the manuscript and would like to resubmit it for your consideration. The modifications were marked are marked in red font in this manuscript. Here are the point-by-point responses:

This study presents intriguing findings on the relationship between plant-based diets and glioma risk in the Chinese population. However, methodological concerns and potential biases limit the generalizability and impact of the results. My concerns and suggestions are outlined below:

  1. The study employs a case-control design which, while appropriate for rare diseases, inherently has limitations. Such a study design is susceptible to both recall and selection biases. The authors acknowledged this limitation, but the possibility of unaddressed confounding factors remains a concern.

=> Responses: Thank you for your suggestion. Indeed, for this study, we cannot avoid recall bias and selection bias. Corresponding measures can only be taken during the research process to minimize the impact of relevant biases. For example, the majority of the case group came from newly diagnosed glioma patients, while the control group came from healthy individuals in the community rather than hospitals. For confounding factors, as the current risk factors for glioma were not yet clear, only the therapeutic dose of ionizing radiation was considered an influencing factor. In this study, we have considered potential confounding factors as much as possible, such as allergies, head injuries, occupation, physical activity, etc., and have also obtained results similar to the overall population in subgroup analysis. Please refer to Line 327-329 in the manuscript.

“First, this study was a case-control study, and it was important to note that there may be inherent recall bias and selection bias associated with this study design, which may not be entirely avoidable. However, due to the extremely low incidence of glioma, case-control studies were still the primary type of study for investigating the causes of glioma.”

  1. Using a convenience sampling method for selecting the case group raises concerns about the representativeness of the sample. Such a method can introduce bias and limit the external validity of the findings.

=> Responses: Thank you for your suggestion. Convenient sampling did indeed bring potential selection bias. However, due to the extremely low incidence rate of glioma, there are few cohort studies on glioma or other brain tumors, so only case-control studies can be used for preliminary exploration, while hospital-based case-control studies cannot avoid convenient sampling. We have added corresponding deficiencies in the limitations of our discussion, with the hope of improving them in future research. Please refer to Line 331-332 in the manuscript.

“Therefore, we look forward to using prospective cohort studies in future studies to further validate the relationship between the two.”

  1. While Beijing Tiantan Hospital may be authoritative in the field of neuro-oncology in China, sourcing all glioma patients from a single hospital can limit the generalizability of the study findings.

=> Responses: Thank you for your suggestion. There is indeed a selection bias effect in single center case-control studies. However, the incidence rate of glioma is extremely low (5-8/100000), which severely limited the development of multi center case-control studies. At the beginning of the study, we also attempted to collect glioma patients from multiple hospitals. However, in the preliminary experiment, other hospitals of the same level rarely diagnosed and treated glioma patients. In the preliminary experiment, the number of glioma patients in other hospitals was less than 5% of Tiantan Hospital, which was considered to be related to the complexity and difficulty of glioma diagnosis and treatment. Due to the fact that the cases collected in this study are not limited to the Beijing area, but come from various parts of China, the corresponding bias has been reduced to some extent. Therefore, we have added relevant limitations in the discussion section. Please refer to Line 335-339 in the manuscript.

“Secondly, the glioma patients in this study were mainly from Beijing Tiantan Hospital, which may have some selection bias. However, Beijing Tiantan Hospital is currently the most authoritative medical institution in the field of neuro-oncology in China. Its patients also come from all over the country, so it was reasonably representative of the Chinese population.”

  1. Relying solely on food frequency questionnaires (FFQs) can lead to errors in dietary recall and may not capture the intricacies of participants' diets. Moreover, FFQ validation is mentioned, but details on the process or its accuracy are lacking.

=> Responses: Thank you for your suggestion. Due to the fact that the food frequency questionnaire is the main method for evaluating long-term dietary intake, we chose this method in our study. The validation of the food frequency questionnaire has been elaborated in detail in previous studies, so this part of the content has not been described in detail in the manuscript. Overall, in order to further validate the reproducibility and validity of the questionnaire in this study, approximately one year later, we conducted a further survey of 30 healthy controls, collected dietary information from participants through a food frequency questionnaire and a 24-hour recall (two working days and one rest day), calculated food consumption and nutritional intake, and evaluated the reproducibility and validity of the questionnaire through means and correlation coefficients. To improve repeatability, the correlation coefficients for the food group were 0.502 to 0.847, and the correlation coefficients for the nutritional group were 0.437 to 0.807. In terms of validity, the correlation coefficients for the food group were 0.381~0.779, and the correlation coefficients for the nutritional group were 0.380~0.804. Therefore, the reproducibility and validity of this food frequency questionnaire were good [1].

  1. Zhang, W.; He, Y.; Kang, X.; etc. Association between dietary minerals and glioma: A case-control study based on Chinese population. Front Nutr 2023, 10, 1118997, doi:10.3389/fnut.2023.1118997.

5. Although the study controlled for some potential confounders, it did not account for others that could be crucial, such as genetic factors, environmental exposures, or other lifestyle factors that may influence glioma risk.

=> Responses: Thank you for your suggestion. At present, the risk factors for glioma are not clear, and only the therapeutic dose of ionizing radiation is considered a risk factor. In this study, populations with other cancers and occupations that may be exposed to high-dose radiation had been excluded. For genetic factors, out of the 506 patients included in this study, only one patient complained that their brother had also suffered from glioma, while the other patients reported that their immediate family members had not suffered from glioma or other brain tumors within three generations. For environmental factors, there are currently no other environmental factors reported to be clearly related to glioma, but we have considered common risk factors for tumors, such as smoking and non-ionizing radiation high-risk areas reported in previous literature. In terms of current research on the etiology of glioma, this study has considered almost all potential factors affecting glioma.

  1. The authors make broad claims regarding the association between plant-based diets and glioma risk based on a sample exclusively from the Chinese population. Generalizing these findings to other populations can be misleading.

=> Responses: Thank you for your suggestion. This was our mistake. Previous studies have explored the association between plant-based diet and glioma in the Iranian population, considering that the Chinese diet is predominantly plant-based, similar studies have been conducted to verify the reproducibility of the association. This study is limited to the Chinese population and should not be extrapolated to a wider population. We have made corresponding supplements in the limitations section of the discussion. Please refer to Line 339-341 in the manuscript.

“Considering that sample was only from the China population, its representativeness was limited, and its conclusions should be extrapolated to other populations with caution.”

  1. The lack of significant association between the healthy plant-based diet index (hPDI) and glioma risk is puzzling, especially given the observed effects of the overall PDI and uPDI. This inconsistency is not adequately addressed or explained.

=> Responses: Thank you for your suggestion. We currently believe that the inconsistency between hPDI and PDI results was related to tubers. The hPDI only considered healthy plant-based foods such as vegetables and fruits, while ignoring tubers in unhealthy plant-based foods. In the results of food group analysis, there was a significant negative correlation between tubers and glioma. Potatoes have a positive score in both PDI and uPDI, while they have a negative score in hPDI, resulting in inconsistent results. Due to the high content of anti-glioma active ingredients (chlorogenic acid, etc) in tuber foods such as potatoes [2,3], the intake of tubers varies greatly between the Iranian and Chinese populations, which may also lead to different results in related plant-based dietary indices. Therefore, a plant-based diet was a beneficial combination of healthy plant foods and tubers for glioma. Please refer to Line 324-326 in the manuscript.

“In addition, tubers such as potatoes are rich in chlorogenic acid [51], which has been found to have potential therapeutic effects on glioma [52]. This may also lead to tubers being a positive food for glioma, rather than a negative one.”

  1. Cebulak, T.; Krochmal-Marczak, B.; Stryjecka, M.; etc. Phenolic Acid Content and Antioxidant Properties of Edible Potato (Solanum tuberosum L.) with Various Tuber Flesh Colours. Foods 2022, 12, doi:10.3390/foods12010100.
  2. Kang, Z.; Li, S.; Kang, X.; etc. Phase I study of chlorogenic acid injection for recurrent high-grade glioma with long-term follow-up. Cancer Biol Med 2023, 20, 465-476, doi:10.20892/j.issn.2095-3941.2022.0762.

  1. The protective effects of tubers are highlighted, but it's unclear why they were initially classified as unhealthy plant foods in the PDI. This discrepancy necessitates clearer justification.

=> Responses: Thank you for your suggestion. The research and development of plant diet index originally came from exploring the relationship between dietary patterns and diabetes. One of the main nutrients of potato was carbohydrate, so it was considered as an unhealthy food for diabetes [4]. However, this effect on glioma may not be consistent with that of diabetes. Due to the high content of anti-glioma active ingredients (chlorogenic acid, etc) in tuber foods such as potatoes [2,3], we conducted an analysis of the correlation between food components and glioma, and the results suggested that tubers may be beneficial for gliomas

  1. Cebulak, T.; Krochmal-Marczak, B.; Stryjecka, M.; etc. Phenolic Acid Content and Antioxidant Properties of Edible Potato (Solanum tuberosum L.) with Various Tuber Flesh Colours. Foods 2022, 12, doi:10.3390/foods12010100.
  2. Kang, Z.; Li, S.; Kang, X.; etc. Phase I study of chlorogenic acid injection for recurrent high-grade glioma with long-term follow-up. Cancer Biol Med 2023, 20, 465-476, doi:10.20892/j.issn.2095-3941.2022.0762.
  3. Satija, A.; Bhupathiraju, S.N.; Rimm, E.B.; etc. Plant-Based Dietary Patterns and Incidence of Type 2 Diabetes in US Men and Women: Results from Three Prospective Cohort Studies. Plos Med 2016, 13, e1002039, doi:10.1371/journal.pmed.1002039.

9.  The paper frequently compares its findings to those from studies in other populations, particularly Iranian studies. While drawing parallels can be informative, the paper should stand on its own merit.

=> Responses: Thank you for your suggestion. We further compared it with Iranian study in the discussion section. In contrast, this study not only explored the association between glioma and plant-based diet, but also supplemented other evidence, including: 1) the association between glioma with different pathological subtypes and grades and plant-based diet; 2) The dose-response relationship between plant-based dietary index and glioma risk; 3) The association between some food groups and glioma.

  1. While potential mechanisms linking plant-based diets and glioma are proposed, they are speculative. A more thorough exploration and stronger evidence are required before suggesting any causative relationship.

=> Responses: Thank you for your suggestion. This is a very good suggestion! Our study actually focused on population epidemiology, exploring for the first time the association between plant-based diet and glioma in the Chinese population, in order to identify risk factors for glioma from a dietary perspective and provide a basis for primary prevention of glioma. However, as plant-based diet was a broad concept, it was difficult to conduct experiments like drugs or other intervention factors. Therefore, its related mechanisms need further exploration. However, our research team has conducted research on the mechanism of related phytochemicals in the intervention and treatment of brain tumors based on plant-based diets [3, 5-8], providing some ideas for this. We have added this section to the limitations of the discussion. Please refer to Line 341-344 in the manuscript.

“In addition, due to the complex composition of plant-based diets, the vast majority of research on tumors was mainly focused on population studies. Therefore, the an-ti-tumor mechanism of plant-based diets was not yet clear, and further exploration was needed in future studies.”

  1. Kang, Z.; Li, S.; Kang, X.; etc. Phase I study of chlorogenic acid injection for recurrent high-grade glioma with long-term follow-up. Cancer Biol Med 2023, 20, 465-476, doi:10.20892/j.issn.2095-3941.2022.0762.
  2. Wang, C.; Cai, Z.; Huang, Y.; etc. Honokiol in glioblastoma recurrence: a case report. Front Neurol 2023, 14, 1172860, doi:10.3389/fneur.2023.1172860.
  3. Li, S.; Chen, J.; Fan, Y.; etc. Liposomal Honokiol induces ROS-mediated apoptosis via regulation of ERK/p38-MAPK signaling and autophagic inhibition in human medulloblastoma. Signal Transduct Tar 2022, 7, 49, doi:10.1038/s41392-021-00869-w.
  4. Li, S.; Li, L.; Chen, J.; etc. Liposomal honokiol inhibits glioblastoma growth through regulating macrophage polarization. Ann Transl Med 2021, 9, 1644, doi:10.21037/atm-21-1836.
  5. Zhang, W.; Wang, C.; Chen, F.; etc. Phytochemicals and Glioma: Results from Dietary Mixed Exposure. Brain Sci 2023, 13, doi:10.3390/brainsci13060902.

Round 2

Reviewer 1 Report

The revised manuscript has mentioned the comments.  

Author Response

Dear Reviewer

The authors would like to express their gratitude for the positive and constructive comments and suggestions. We have revised the manuscript and would like to resubmit it for your consideration. The modifications were marked are marked in red font in this manuscript. Here are the point-by-point responses:

  1. The revised manuscript has mentioned the comments.

=> Responses: Thank you for your suggestion. We have supplemented the results of the subgroup analysis in the main text of the manuscript. Please refer to Line 208-214 in the manuscript.

“The results showed that for sex subgroups, smoking status subgroups, BMI subgroups, family history of cancer subgroups, and other subgroups, uPDI was associated with an increased risk of glioma, while hPDI was not associated with glioma risk, which were consistent with the overall population results. However, for PDI, the results were not significant in the low age group, low BMI subgroup, low household income group, and allergic population, which was inconsistent with the overall population results and may be related to the small sample size of some subgroups (Table S1).”

Reviewer 4 Report

The authors should discuss is there a statistically significant difference between Model 1 and Model 2 based on Table S1. The results of subgroup analysis.

Section 5 Conclusions is very brief. I suggest to extend or merge with Section 4 Discussion.

Author Response

Dear Reviewer

The authors would like to express their gratitude for the positive and constructive comments and suggestions. We have revised the manuscript and would like to resubmit it for your consideration. The modifications were marked are marked in red font in this manuscript. Here are the point-by-point responses:

1. The authors should discuss is there a statistically significant difference between Model 1 and Model 2 based on Table S1. The results of subgroup analysis.

=> Responses: Thank you for your suggestion. We have supplemented the results of the subgroup analysis in the main text of the manuscript. Please refer to Line 208-214 and Line 253-260 in the manuscript.

“The results showed that for sex subgroups, smoking status subgroups, BMI subgroups, family history of cancer subgroups, and other subgroups, uPDI was associated with an increased risk of glioma, while hPDI was not associated with glioma risk, which were consistent with the overall population results. However, for PDI, the results were not significant in the low age group, low BMI subgroup, low household income group, and allergic population, which was inconsistent with the overall population results and may be related to the small sample size of some subgroups (Table S1).”

“In addition, we conducted sufficient subgroup analysis based on relevant confounding variables. For uPDI and hPDI, the results showed that the majority of subgroup results were consistent with the overall population, but the PDI results were not significant in individual subgroups, which was considered to be related to the small sample size of individual subgroups. However, the significance of Model 1 and Model 2 results for the vast majority of subgroups was basically consistent. However, for PDI, in the elderly subgroup and high BMI subgroup, Model 1 was significant while Model 2 was not, indicating that there may be potential confounding factors that may affect the results.”

2. Section 5 Conclusions is very brief. I suggest to extend or merge with Section 4 Discussion.

=> Responses: Thank you for your suggestion. We combined the conclusion with the discussion. Please refer to Line 363-366 in the manuscript.